# A Harmonious Satellite-Unmanned Aerial Vehicle-Ground Measurement Inversion Method for Monitoring Salinity in Coastal Saline Soil

**Suming Zhang and Gengxing Zhao \***

National Engineering Laboratory for Efficient Utilization of Soil and Fertilizer Resources, College of Resources and Environment, Shandong Agricultural University, Taian 271018, China
\* Correspondence: zhaogx@sdau.edu.cn; Tel.: +86-133-4528-3157

**Abstract:** Soil salinization adversely impacts crop growth and production, especially in coastal areas which experience serious soil salinization. Therefore, rapid and accurate monitoring of the salinity and distribution of coastal saline soil is crucial. Representative areas of the Yellow River Delta (YRD)—the Hekou District (the core test area with 140 sampling points) and the Kenli District (the verification area with 69 sampling points)—were investigated. Ground measurement data, unmanned aerial vehicle (UAV) multispectral imagery and Sentinel-2A multispectral imagery were used as the data sources and a satellite-UAV-ground integrated inversion of the coastal soil salinity was performed. Correlation analyses and multiple regression methods were used to construct an accurate model. Then, a UAV-based inversion model was applied to the satellite imagery with reflectance normalization. Finally, the spatial and temporal universality of the UAV-based inversion model was verified and the soil salinity inversion results were obtained. The results showed that the green, red, red-edge and near-infrared bands were significantly correlated with soil salinity and the spectral parameters significantly improved this correlation; hence, the model is more effective upon combining spectral parameters with sensitive bands, with modeling precision and verification precision of the best model being 0.743 and 0.809, respectively. The reflectance normalization yielded good results. These findings proved that applying the UAV-based model to reflectance normalized Sentinel-2A images produces results that are consistent with the actual situation. Moreover, the inversion results effectively reflect the distributions characteristic of the soil salinity in the core test area and the study area. This study integrated the advantages of satellite, UAV and ground methods and then proposed a method for the inversion of the salinity of coastal saline soils at different scales, which is of great value for real-time, rapid and accurate soil salinity monitoring applications.

**Keywords:** remote sensing; inversion; UAV; Sentinel-2A satellite; soil salinity

---

## 1. Introduction

Land resources worldwide are being irreversibly reduced and degraded because of anthropogenic pressures, variations in land use patterns and climate change [1,2]. An important example is soil salinization, which constitutes one of the most widespread soil degradation processes on Earth [3,4]. The coastal saline soil of the Yellow River Delta (YRD) is an important land resource in China and as a result of various factors, saline soil is broadly distributed throughout the YRD with a salt content that changes frequently [5]. These conditions negatively impact the regional agricultural sustainability and environmental health and may trigger severe losses of soil productivity and, ultimately, cause desertification [6–9]. As a result, to ensure the proper utilization of land resources and to protect the ecological environment, it is urgently necessary to monitor soil salinity (SS) in the YRD in real time [10].

Traditional soil salinity monitoring methods involve the acquisition of field measurements. However, with their introduction in the 1990s, remote sensing and geographic information system (GIS) technologies initiated a new era in which soil salinity could be dynamically monitored on a large scale [11–15]. Because satellite platforms can provide massive quantities of information over large spatial areas at low cost and at frequent intervals, satellite remote sensing has gradually replaced traditional soil salinity monitoring methods, which are inefficient and expensive [16]. However, satellites exhibit various disadvantages, such as fixed orbits and long revisit periods and satellite data suffer from atmospheric effects, especially low spatial resolution [17]; thus, it is difficult to perform high-precision, real-time inversions in the field.

In recent years, the technology of unmanned aerial vehicles (UAVs) has gradually been integrated into the civil field, becoming a popular subject of practical agricultural research and applications [18]. Compared with traditional measurement methods, UAV remote sensing provides a nondestructive and cost-effective approach for rapid soil salinity monitoring. Compared with other remote sensing platforms (i.e., satellites), UAV platforms are easier to build, can fly at lower altitudes and over different types of areas and can capture images with high spatial and temporal resolution [19,20]. Consequently, UAV technology has been widely used in agriculture [21–25]. However, the current UAV technology still exhibits some limitations. For example, compared with satellites, UAVs are unsuitable for the acquisition of imagery on a large scale [26]. In addition, UAVs are not permitted in certain areas due to privacy concerns. Therefore, although an inversion model base on UAV imagery boasts a higher accuracy, satellite remote sensing remains the best source of basic imagery when acquiring information over a large region.

Scholars have conducted research on the application of multisource data and generated good results [27–29]. However, most data sources are the synergies of radar and optical data and the combination of various satellite remote sensing data. Most of the research objects focus on vegetation canopy parameters. Accordingly, the harmonious use of satellite, UAV and ground data should be rare and feasible and is thus worthy of further research.

In this study, considering the factors discussed above, representative regions of the YRD—namely, the Hekou District and Kenli District—which possess coastal saline soil were chosen as the study areas in which to perform the following: (1) construct inversion models of the soil salinity based on UAV imagery and field-measured data and select the best model; (2) use the relationship between the reflectance of UAV and satellite images to normalize the reflectance of satellite images; and (3) apply the best inversion model to the normalized satellite imagery to achieve scaled-up soil salinity inversions.

The advantages of ground-measured, UAV and satellite methods were fully harmonized in this study. In addition, the scale, precision and spatial-temporal resolutions of soil salinity inversions were improved.

## 2. Data and Methods

The data sources in this paper include satellite, UAV and ground-measured data. When building the UAV-based model, a ground sampling data point corresponds to a pixel of UAV imagery. When building the satellite-based model, a ground sampling point data corresponds to a pixel of satellite imagery. The method for building the satellite-based model is the same as that used for the UAV-based model, as shown in Section 2.2. When calculating the normalization coefficient of reflectivity of the satellite images, there are approximately 1600 UAV image pixels corresponding to a satellite image pixel of the same area.

## 2.1. Study Area

The study area is located in the representative region of the YRD, namely, the Hekou District (37°45′~38°10′ N, 118°07′~119°05′ E) and the Kenli District (37°24′~38°10′ N, 118°15′~119°19′ E). The study area exhibits a warm temperate continental monsoon climate that is dry and windy in spring [30]. Evaporation in spring far exceeds precipitation, with an evaporation-precipitation ratio of 7.6; thus, the vegetation coverage in the study area is low, and the soil is subjected to serious seasonal salt return and salt accumulation. The terrain of the study area slopes slightly from the southwest to the northeast. Additionally, the groundwater table is shallow and highly mineralized. The main land use types are agricultural land and unused land and the main soil type is coastal (tidal) saline soil with a light texture and strong capillary action. Due to the above mentioned factors, the degree of soil salinization in the study area is high and saline soil is widely distributed [31,32]. These conditions seriously affect the development and utilization of regional land resources and the sustainable development of the social economy [33].

A farm of the "Bohai Granary" project located in the Hekou District was selected as the core test area, in which field measurements were performed and a UAV flight test was carried out. The differences in the soil salt content throughout the core test area are obvious, and the field contains all salinization grades; thus, this field served as the foundation for the universality of the soil salinity inversion model. In addition, the Kenli District was chosen as the verification area to validate the inversion results from the UAV-based model on the satellite image after reflectance normalization (Figure 1).

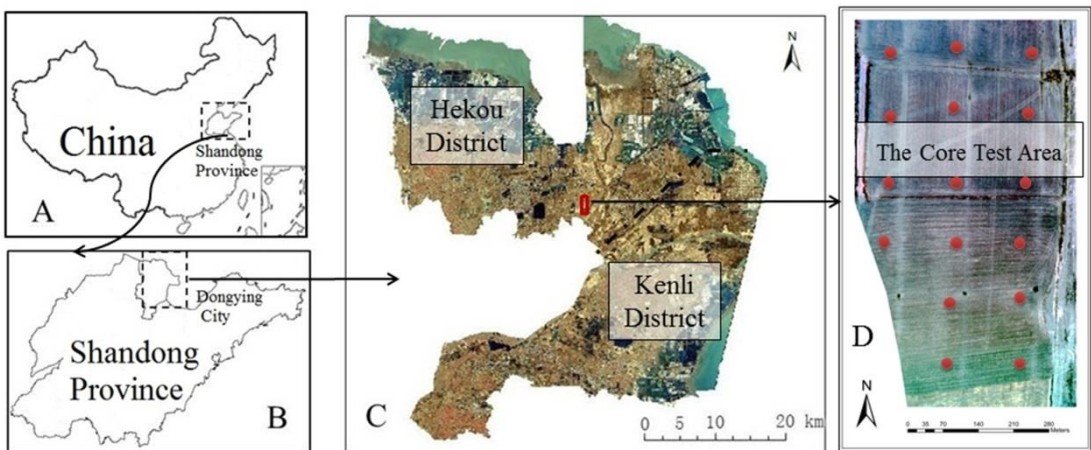

**Figure 1.** Location of the study area (**A**) China; (**B**) Shandong Province; (**C**) the study area; and (**D**) part of the core test area, with the red dots representing the sampling points.

## 2.2. Data Acquisition and Preprocessing

### 2.2.1. Acquisition of Soil Salinity Data

In the core test area, 160 sampling points were established. The sampling points were selected based on the following factors: (1) the centers of observation areas that were spatially representative with identical land use types were taken as sampling points; (2) an observation area of 20 m × 20 m was covered; and (3) the crop type, crop growth and vegetation coverage in the agricultural land were all uniform. An EC110 portable salinity meter equipped with a 2225FS T series probe (in which the temperature correction for the electrical conductivity had already been completed) (Spectrum Technologies, Inc., USA) was used to make 5 measurements at and near the sampling points, with a range of no more than 5 cm × 5 cm. The measured value was the soil electrical conductivity (EC) in units of dS/m or mS/cm. The coordinates (longitude and latitude) of the sampling points were measured with a Trimble GEO 7X centimeter handheld differential GPS (DGPS)(Trimble inc. Sunnyvale, California,

USA) with an accuracy of approximately 3~25 mm. Moreover, the coordinates of ten representative ground objects around the core test area were recorded as the ground control points (GCPs) during geometric correction. The sampling was conducted on 5–10 March 2018.

The soil salinity was determined from the regression equation derived by predecessors, $SS = 2.18 \times EC + 0.727$ [34], where SS represents the soil salinity in units of g/kg. The average soil salinity after repeated measurements was taken as the measured soil salinity at each sampling point.

In total, 69 sampling points were set up based on a random distribution of land use types in the whole validation area. The measured soil salinity values were significantly different, with different salinity degrees range from 1.34 g/kg to 9.52 g/kg. The factors governing the selection of the sampling area and the method of investigation were the same as those employed in the core test area. The survey was performed on the 25th to 27th April 2016.

### 2.2.2. Acquisition and Processing of UAV Imagery

A Sequoia multispectral camera was mounted on a Dajiang Matrice 600 Pro UAV (loaded mass: 5.5 kg; flying time: 16 min) ( SZ DJI Technology Co.,Ltd. Shenzhen, Guangdong province, China). The camera captures four bands: green, red, red-edge and near-infrared (Table 1) [35]. The UAV test was conducted from 13:00 to 14:00 each day from the 7th to 9th March 2018, in the core test area to obtain UAV imagery. The UAV hovered at a height of 100 m, the flight speed was 5 m/s, the image acquisition interval was set to 1.5 s and the area covered was approximately 1 km$^2$. During the process of taking photographs, the sunshine sensor equipped with the Sequoia could correct the illumination difference to calibrate the intrinsic radiation. By taking the whiteboard images by the Sequoia camera and loading the whiteboard images and whiteboard parameters corresponding to each band in the Pix4D software (Pix4D S.A. Route de Renens 241008 Prilly, Switzerland) before image mosaicking, extrinsic radiation calibration and spectral calibration were achieved.

**Table 1.** Consistency of model Sentinel-2A and unmanned aerial vehicle (UAV) data.

| Name of Bands | Sentinel-2A Data | | Sequoia (UAV) Data | | |
|---|---|---|---|---|---|
| | Bands | Central Wavelength (nm) | Bands | Central Wavelength (nm) | FWHM (nm) |
| Bg | B3-Green | 560 | Green | 550 | 20 |
| Br | B4-Red | 665 | Red | 660 | 20 |
| Breg | B6-Vegetation Red Edge | 740 | Red Edge | 735 | 5 |
| Bnir | B7-Vegetation Red Edge | 783 | Near IR | 790 | 20 |

Note: FWHM (Full Wave at Half Maximum)

Pix4D software was used to preprocess the UAV imagery, which included operations such as mosaicking, converting the data into surface reflectance and extrinsic radiation calibration. Finally, a multispectral orthophoto of the core test area was obtained with a resolution of 4~5 cm (Figure 1D).

### 2.2.3. Acquisition and Processing of Sentinel-2A Satellite Data

The Sentinel series of satellites was launched by the European Space Agency (ESA) for the Copernicus Programme [36]. Within the Sentinel series, Sentinel-2 consists of two optical satellites, namely, Sentinel-2A and Sentinel-2B. In this paper, the Sentinel-2A products were downloaded from the ESA Copernicus data sharing website (https://scihub.copernicus.eu/). We selected images covering the study area that were acquired at the same time on the 1st May 2016, and the 1st March 2018.

The downloaded data are atmospheric reflectance data that have been geometrically corrected. Using the Sentinel Application Platform (SNAP) software provided by the ESA, atmospheric corrections and resampling were applied to the data and the data were exported in an ENVI format. Then, the software ENVI 5.1 (Exelis Visual Information Solutions company, USA) was employed to mosaic the data, extract the reflectivity, perform image clipping, inversion, and classification and then to output the image.

Sentinel-2A provides multispectral data with 13 bands. Considering the consistency between the Sentinel-2A and UAV data and the requirements for building a soil salinity model, the satellite bands that are consistent with the wavelength range of the Sequoia camera were selected, as shown in Table 1.

### 2.3. Soil Salinity Inversion Model Based on UAV Imagery

Soil salinity inversion models were constructed by analyzing the field-measured soil salinity data and UAV imagery. The ground sampling points and pixels of UAV imagery share a one-to-one correspondence, and they were divided into a modeling set and a validation set at a ratio of 2:1.

First, a relevance analysis between all field-measured and UAV data was performed, after which the sensitive soil salinity bands were screened. Second, new spectral parameters, including multiband information, were generated via band combination operations (i.e., adding, subtracting, multiplying and dividing operations between bands). Then, the spectral parameters were screened. The indicator of the screening was the correlation coefficient (expressed as R) and a band or band combination with a larger absolute value of R was screened as a sensitive band or spectral parameter, which showed a higher correlation with the soil salinity [37]. In this paper, the commonly used Pearson correlation coefficient is adopted.

Finally, the sensitive bands and spectral parameters of the modeling set were used as independent variables and the soil salinity was used as a dependent variable to construct the soil salinity inversion model by a variety of regression methods.

The validation data set was used for model validation. The spectral data of each sampling point in the verification set were incorporated into the models to determine the soil salinity inversion values of each sampling point, and a correlation analysis was carried out with the measured values to determine the verification precision.

The modeling precisions and verification precisions (coefficient of determination, $R^2$) were determined to select the best inverse model. $R^2$ was used to compare and evaluate the performance of the models. Table 2 shows the prediction abilities of models with $R^2$ values in different ranges [38]. Relevance analysis and model construction were implemented using Statistical Product and Service Solutions (SPSS 22) software (International Business Machines Corporation, Armonk, New York, USA).

**Table 2.** The prediction abilities of models with $R^2$ values in different ranges.

| Range of $R^2$ | Prediction of Models |
|---|---|
| 0.50~0.65 | Poor |
| 0.66~0.81 | Approximate |
| 0.82~0.90 | Good |
| 0.91~1.00 | Accurate |

### 2.4. Reflectance Normalization of Sentinel-2A Imagery

The average reflectivity of each band of the 1600 pixels of UAV images and the corresponding reflectivity of one pixel of Sentinel-2A imagery were determined and the relationship between them was analyzed.

To determine whether it is feasible to normalize the reflectivity of Sentinel-2A images based on UAV images, the average reflectivity of 140 sampling points in each band of the UAV and Sentinel-2A images was calculated, and the variation trends of the two images were compared. Furthermore, scatter plots of the average reflectivity of the corresponding bands of the UAV and the Sentinel-2A images were generated to prove the correlation between the reflectivities of the two images.

To normalize the reflectance of Sentinel-2A, the mean of ratios correction method was used [39,40]. First, the ratio between the reflectance of the corresponding pixels in the green band of the Sentinel-2A images and the green band of the UAV images was calculated (such as B3 – Green/Green) and then the mean of all ratios was taken as the reflectivity normalization coefficient of the green band. The reflectance normalization coefficients of the other bands were calculated via the same method.

To obtain the normalized Sentinel-2A imagery, the reflectance in each band of the Sentinel-2A imagery was divided by the reflectivity normalization coefficients. This process was implemented in ENVI 5.1.

## 2.5. Validation of the Soil Salinity Inversion Model

The validation of the UAV-based best soil salinity inversion model obtained from Section 2.2 was mainly reflected in the following two aspects.

To verify whether reflectance normalization improves the inversion precision of Sentinel-2A imagery, the precisions under different conditions were compared and analyzed by applying a UAV-based inversion model to the Sentinel-2A imagery before reflectance normalization and after reflectance normalization and applying a Sentinel-2A-based model to the Sentinel-2A imagery. Thus, the feasibility of applying reflectance normalization to the Sentinel-2A imagery was verified.

To verify the spatial and temporal universality of the model, the inversion and interpolation results of the verification area were compared. The Sentinel-2A imagery of the verification area (Kenli District) acquired on the 1st May 2016, was pre-processed and reflectance normalized and the best inversion model was then used to obtain the inversion result. The inversion result was the soil salinity of each pixel in the Sentinel-2A images. Using the field-measured soil salinity data in 2016, the interpolation chart of the soil salinity in the verification area was obtained. Subsequently, the pixels of the inversion result and interpolation chart were classified according to the following criteria: non-salinization (<1 g/kg), mild salinization (1–2 g/kg), moderate salinization (2–4 g/kg), severe salinization (4–6 g/kg) and saline soil (>6 g/kg). A map of the spatial distribution of the soil salinity and a table of the areal proportion of each salinity grade were obtained in this manner.

Inversion was performed using ENVI 5.1 and classification of the pixels was performed using the decision tree method in ENVI 5.1. An interpolation map was produced by the kriging interpolation method of ArcGIS 10.1 (Environmental Systems Research Institute, Inc., California, USA), which is applicable to a relatively small number of sampling points and continuous variables (soil salinities are continuous variables).

## 2.6. Inversion of Soil Salinity

The inversion of soil salinity includes the field scale (the core test area) and the regional scale (the study area). Based on the best inversion model and the UAV imagery obtained in March 2018, a soil salinity inversion map at the field scale was obtained. Based on the best inversion model and the Sentinel-2A imagery (1 March 2018) after preprocessing and reflectance normalization, a scaled-up soil salinity inversion for the whole study area was performed. The inversion results were divided into five grades according to the criteria and method described above and maps of the spatial distribution of soil salinity levels and area statistics for each salinity level were obtained.

## 3. Results and Analysis

### 3.1. Screening of Soil Salinity-Sensitive Bands and Spectral Parameters

#### 3.1.1. Sensitive Bands

The R values between the ground-measured soil salinity and the reflectances in the UAV and Sentinel-2A imagery are shown in Figure 2 (p < 0.01). The reflectance values in the green, red, red-edge and near-infrared bands of the UAV images correspond to Bg, Br, Breg and Bnir, respectively, in the inversion model, and the B3, B4, B6, and B7 bands of Sentinel-2A images correspond to Bg, Br, Breg and Bnir in the inversion model. All bands were significantly correlated with the soil salinity at the 0.01 probability level.

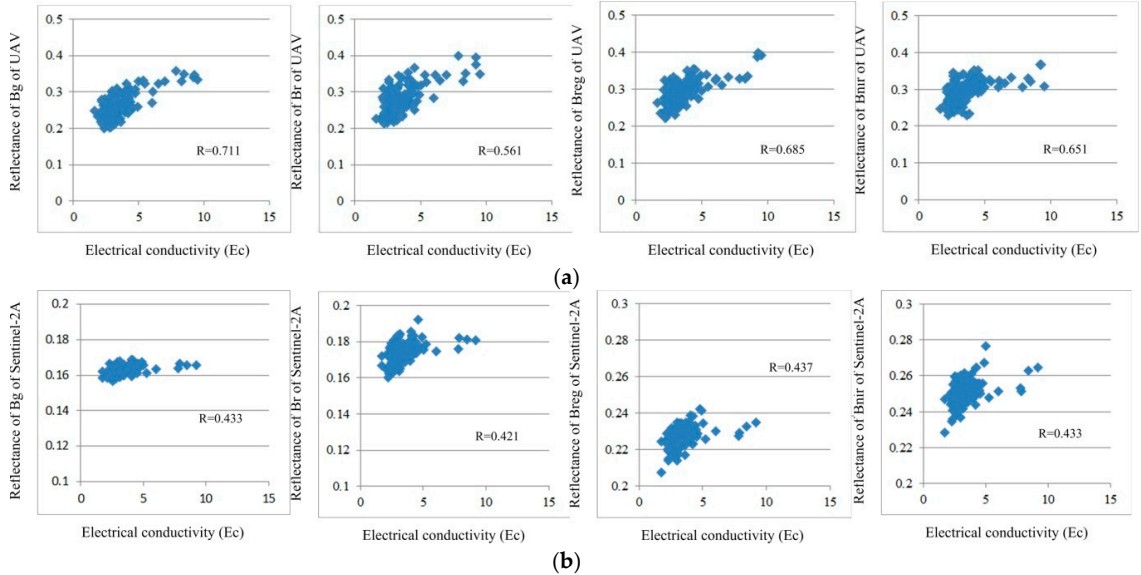

**Figure 2.** Scatter plots between the soil salinity and the spectral reflectances of the UAV (**a**) and Sentinel-2A (**b**) imagery.

The correlation between the reflectivity of Sentinel-2A imagery and the soil salinity is weaker than that of the UAV imagery. Therefore, it is not difficult to infer that the soil salinity inversion model based on UAV imagery is more accurate than the model based on Sentinel-2A imagery. This paper constructed a soil salinity inversion model based on UAV imagery using the Bg, Br, Breg and Bnir bands as the sensitive soil salinity bands.

#### 3.1.2. Spectral Parameters

Figure 3 shows the R values between the spectral parameters based on the UAV imagery and the soil salinity (p < 0.01). Clearly, the correlation between the soil salinity and the reflectance of combined bands increases significantly compared with the correlation between the soil salinity and the reflectance of a single band. The band combinations with R > 0.76 were selected as spectral parameters for the construction of the inversion model.

The correlation of the soil salinity with the band combinations obtained by band addition and multiplication is higher than that with the band combinations obtained by other methods. All spectral parameters contain band Bg, which shows the strongest response to soil salinity.

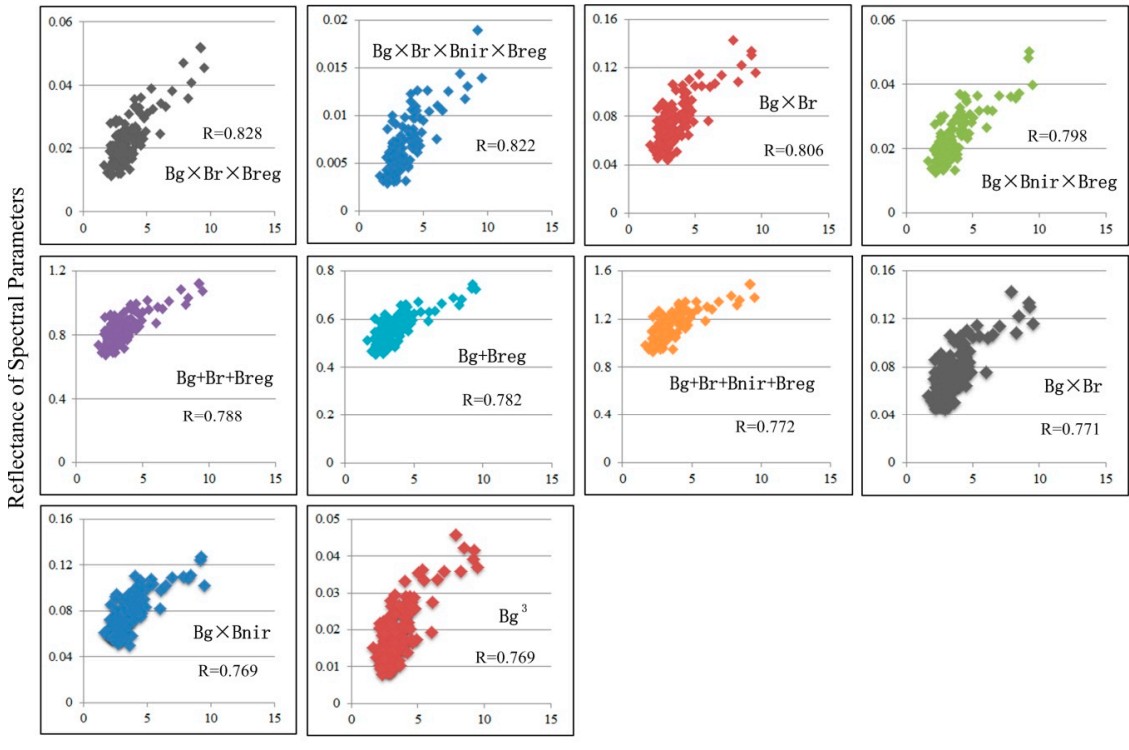

Electrical conductivity (Ec)

**Figure 3.** Scatter plots between the soil salinity and the spectral parameters derived from the UAV imagery.

### 3.2. Soil Salinity Inversion Model Based on UAV Imagery

Based on the UAV imagery, the soil salinity inversion model was constructed by a variety of regression methods with the following three sets of independent variables: the sensitive bands, the spectral parameters, and the sensitive bands and spectral parameters together. The results are shown in Table 3.

**Table 3.** Inversion model of the soil salinity based on UAV imagery.

| Independent Variable | Modeling Method | Independent Variable | Modeling Precision | Verification Precision |
|---|---|---|---|---|
| | | | $R^2$ | $R^2$ |
| Sensitive Bands | Multivariate linear regression | $Y = -5.199 + 17.508 \times Bg + 2.589 \times Br + 11.402 \times Bnir$ | 0.541 | 0.738 |
| Spectral Parameters | Second-order least squares regression | $Y = 8.815 - 20.001 \times Bg + Breg + 16.618 \times (Bg + Br + Bnir + Breg) + 80.798 \times Bg \times Bnir - 390.978 \times Bg \times Br - 427.914 \times Bg \times Breg + 408.278 \times Bg \times Bnir \times Breg + 1885.272 \times (Bg \times Br \times Breg) - 2603.896 \times Bg \times Br \times Bnir \times Breg + 484.547 \times Bg^3$ | 0.724 | 0.76 |
| Sensitive Bands and Spectral Parameters | Multivariate linear regression | $Y = 27.62 - 164.215 \times Bg - 12.307 \times Br - 29.673 \times Bnir - 5.149 \times Breg + 316.529 \times Bg \times Bnir - 134.333 \times Bg \times Bnir \times Breg + 603.149 \times Bg \times Br \times Breg - 1509.075 \times Bg \times Br \times Bnir \times Breg + 415.695 \times Bg^3$ | 0.743 | 0.809 |

In terms of modeling precision, the models constructed using multivariate linear regression or second-order least squares regression are clearly better than the other models and the model constructed using sensitive bands and spectral parameters as independent variables is better than the models constructed using only sensitive bands or spectral parameters as independent variables. The validation precisions of the models are generally high, which shows that the models are all highly applicable and

stable. The best soil salinity inversion model with the highest modeling and verification precisions is
$Y = 27.62 - 164.215 \times Bg - 12.307 \times Br - 29.673 \times Bnir - 5.149 \times Breg + 316.529 \times Bg \times Bnir - 134.333 \times Bg \times Bnir \times Breg + 603.149 \times Bg \times Br \times Breg - 1509.075 \times Bg \times Br \times Bnir \times Breg + 415.695 \times B^3$,
where Y represents the inverted soil salinity values. The model precision of this model is $R^2 = 0.743$,
and the verification precision is $R^2 = 0.809$.

### 3.3. Reflectance Normalization of Sentinel-2A

#### 3.3.1. Comparison of Surface Reflectance between UAV and Sentinel-2A Imagery

Figure 4 compares the surface reflectance of the UAV imagery with that of the Sentinel-2A imagery.
As seen from Figure 4a, the surface reflectance of the UAV imagery is higher than that of the Sentinel-2A
imagery, but their trends are basically the same. The amplitude of the UAV curve is 0.038962, the
amplitude of the Sentinel-2A curve is 0.058313, and the amplitude difference between the two curves is
0.013693. Figure 4b shows a strong correlation between the UAV and satellite imagery, which proves
that it is feasible to normalize Sentinel-2A imagery according to UAV imagery. But the trend line is
further away from 1:1 line.

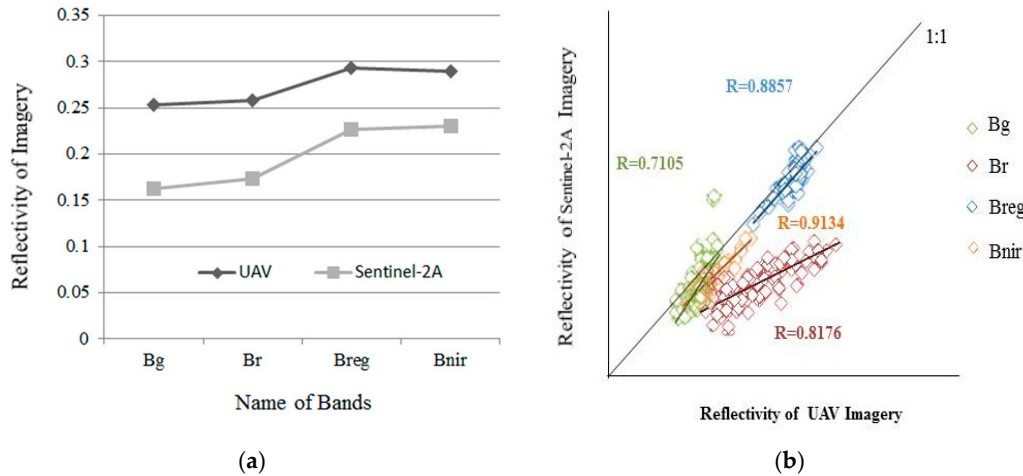

(**a**)　　　　　　　　　　　　　　　　　　　　　　　　　(**b**)

**Figure 4.** Comparison (**a**) and scatter plot (**b**) of the surface reflectances of the UAV and
Sentinel-2A imagery.

#### 3.3.2. Reflectance Normalization Coefficients of Sentinel-2A Imagery

The normalized coefficients of each band were calculated from the reflectivity of the UAV and
Sentinel-2A imagery (Table 4), and the reflectance correlations of the normalized Sentinel-2A and UAV
images are shown in Figure 5.

**Table 4.** Reflectance normalization coefficients of the Sentinel-2A imagery.

| Name of Bands | Bg | Br | Bnir | Breg |
|---|---|---|---|---|
| Reflectance normalization coefficient | 0.640638 | 0.672978 | 0.772553 | 0.796514 |

As shown in Figure 5, the fitting trend lines of Breg and Bnir of the normalized Sentinel-2A and
the corresponding UAV bands are close to the 1:1 line, while those of Bg and Br deviate from this line to
a certain extent. This phenomenon may be affected by the following three factors: first, the difference
between the Sentinel-2A and UAV images in the center wavelength of the Bg band is 10 nm, which is
the largest difference among all the bands; moreover, the gap between Sentinel-2A and UAV in the Bg
band is the largest of all the bands (Figure 4a); and it was hypothesized that the spatial resolution of

the Sentinel-2A image is low, which is affected by the spectral confusion caused by the canopy of only a few crops, and the Bg and Br bands were the most affected.

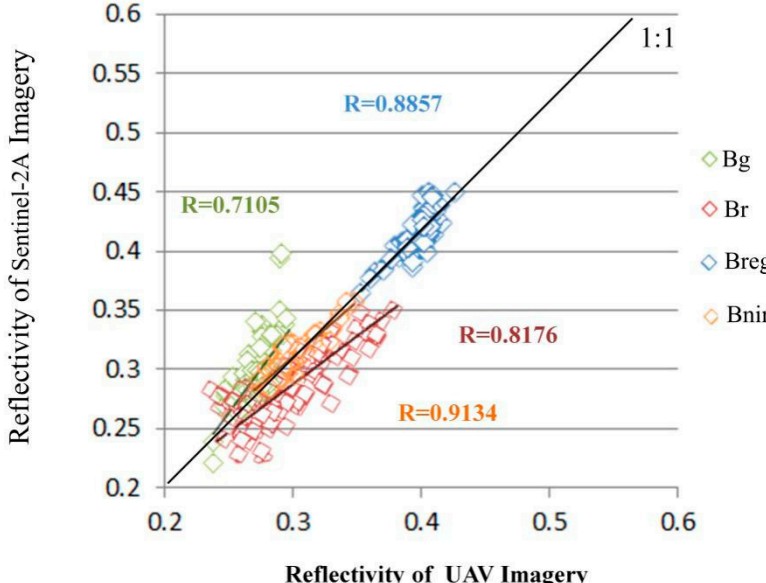

**Figure 5.** The reflectance correlations of the normalized Sentinel-2A and UAV images.

The normalized Breg and Bnir values of Sentinel-2A have high correlations with the corresponding UAV data, reaching 0.8857 and 0.9134, respectively, and show a good correction effect. The normalized Bg and Br values have lower correlations with the corresponding UAV data but both R values are greater than 0.7, which is within the acceptable range. In general, reflectance normalization is an indispensable and effective measure for the scaled-up mapping of soil salinity.

*3.4. Validation of the Best Soil Salinity Inversion Model*

3.4.1. Validation of Reflectance Normalization

Table 5 presents a comparison of the following three cases: (1) the best inversion model based on Sentinel-2A imagery is applied directly to the Sentinel-2A imagery; (2) the best inversion model based on UAV imagery is applied to the Sentinel-2A imagery before reflectance normalization; and (3) the best inversion model based on UAV imagery is applied to the Sentinel-2A imagery after reflectance normalization.

**Table 5.** Comparison of inversion precision before and after reflectance normalization.

| Case | Formula | Modeling Precision | Verification Precision |
|------|---------|-------------------|------------------------|
| | | $R^2$ | $R^2$ |
| (1) | $y = 124.376 - 143.334 \times Bg + 152.463 \times Br + 21.376 \times Bnir + 163.158 \times Breg - 785.155 \times (Bg + Bnir) + 4992.772 \times Bg \times Bnir - 4249.989 \times Bg \times Breg + 3043.255 \times Br \times Breg$ | 0.446 | 0.3 |
| (2) | $y = 27.62 - 164.215 \times Bg - 12.307 \times Br - 29.673 \times Bnir - 5.19 \times Breg + 316.529 \times Bg \times Bnir - 134.333 \times Bg \times Bnir \times Breg + 603.149 \times Bg \times Br \times Breg - 1509.075 \times Bg \times Br \times Bnir \times Breg + 415.695 \times Bg^3$ | 0.602 | 0.317 |
| (3) | The same as case (2) | 0.602 | 0.585 |

Note: y represents the inversion value of the soil salinity.

Regarding the modeling precisions, Table 6 shows that case (3) is much more precise than both case (2) and case (1), which shows that the model based on UAV imagery cannot be applied directly to satellite images without reflectance normalization and that the precision of the model based on satellite imagery is lower than that based on UAV imagery. Regarding the verification precisions, case (3) exhibits the best performance, while case (1) exhibits the lowest, which demonstrates that the stability and universality of satellite-based modeling are low, while those of UAV-based modeling are high; thus, UAV-based models are more suitable for further applications.

**Table 6.** Area statistics of the classification map and interpolation map in 2016 (without tidal flats, water, etc.). Unit: %.

| Soil Salinity Level | Non-saline | Mild Salinization | Moderate Salinization | Severe salinization | Saline Soil |
|---|---|---|---|---|---|
| Inversion | 0 | 3.03 | 41.21 | 12.83 | 42.93 |
| Interpolation | 0 | 3.03 | 41.45 | 12.44 | 43.08 |

In summary, applying the best inversion model based on UAV imagery to Sentinel-2A imagery with reflectance normalization is feasible and achieves the best effect.

### 3.4.2. Inversion and Validation in the Verification Area (Kenli District)

The soil salinity spatial distribution map (a) and the soil salinity interpolation map (b) are shown in Figure 6. The spatial distribution characteristics of soil salinity in Figure 6a are similar to those in Figure 6b, which proves that the spatial accuracy of the inversion is relatively high.

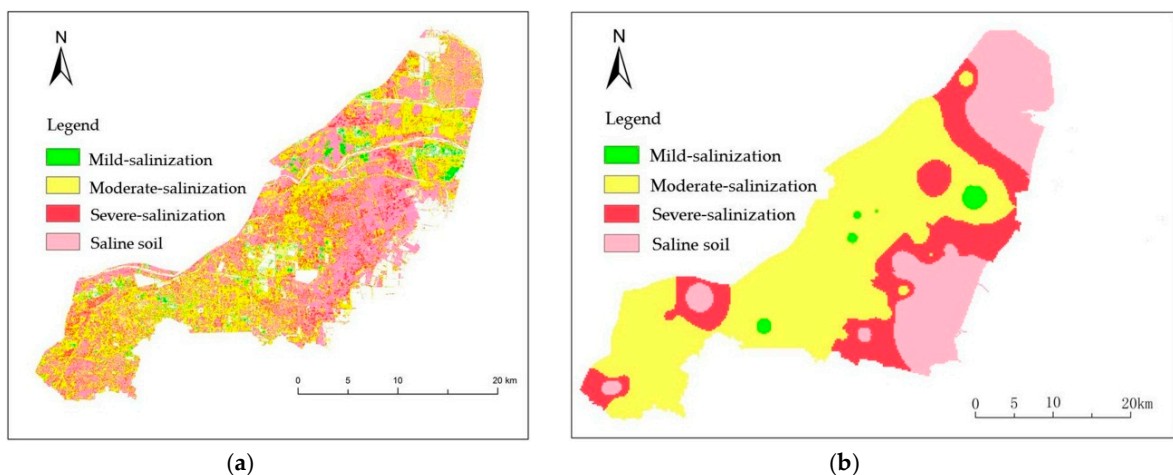

(**a**)　　　　　　　　　　　　　　　　　　　　　　　(**b**)

**Figure 6.** Distribution map (**a**) and interpolation map (**b**) of the soil salinity in the verification area (Kenli District) in 2016.

Table 6 further demonstrates that the inversion results of the area of each soil salinity grade are highly consistent with the spatial interpolation results; thus, the strong temporal and spatial universality of the model has been verified.

### 3.5. Multiscale Inversion of Soil Salinity

### 3.5.1. Field Scale Inversion

Figure 7 presents the soil salinity inversion results in the core test area (0.3 km × 1.2 km). The proportions of soil belonging to the different salinization grades are shown in Table 7.

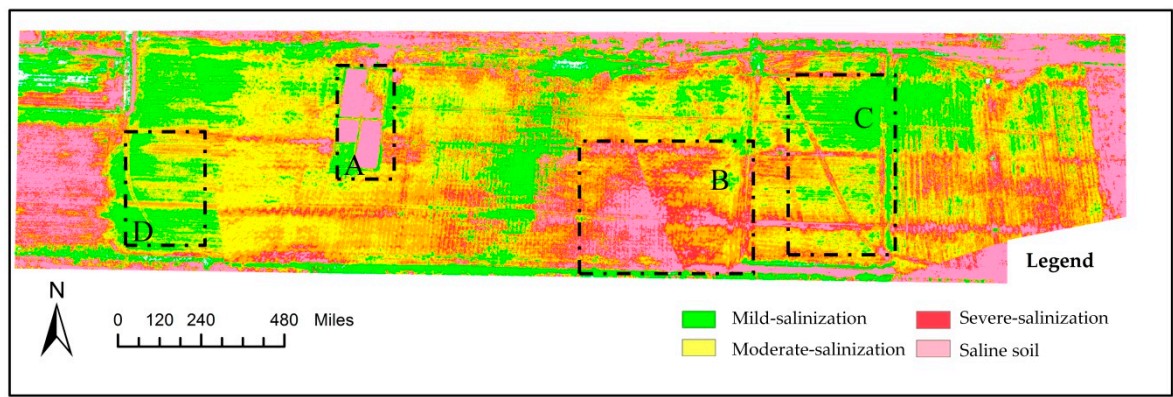

**Figure 7.** Distribution map of the inverted soil salinity in the core test area in 2018.

**Table 7.** Classification results of the core test area in 2018. Unit: %.

| Soil Salinity Level | Non-Saline | Mild Salinization | Moderate Salinization | Severe Salinization | Saline Soil |
|---|---|---|---|---|---|
| Proportion of inversion result | 0 | 18.11 | 27.97 | 24.16 | 29.76 |

Overall, the salinization characteristics in the core test area display a gradually increasing trend from west to east. The proportion of saline soil is the largest, with a value of 29.76%, whereas the proportion of mild salinization is the lowest, which is consistent with the actual situation of the core test area. Due to specific research purposes, the soil salinity in area A is much higher than that in the surrounding soil. Furthermore, the soil salinity of area B is higher due to the lack of irrigation [41,42], although the boundary with the surrounding soil is not obvious. In the areas crushed by wheels (such as tracks in areas C and D), the salinity is increased by one grade relative to the surrounding areas due to compaction and bareness of the soil [43]. According to the description provided above, the model clearly boasts high inversion precision at the field scale and can accurately reflect both the overall trend and the local characteristics of the soil salinity.

3.5.2. Scaled-Up Inversion in the Study Area

Figure 8 illustrates the soil salinity inversion results based on the best inversion model and Sentinel-2A imagery after applying reflectance normalization. The distribution map of the soil salinity levels in the study area in 2018 was obtained, and the areal proportion of each soil salinity grade was calculated accordingly (Table 8).

**Table 8.** Areal statistics of each inverted soil salinity grade in 2018 (without tidal flats, water, etc.); Unit: %.

| Soil Salinity Level | Non-Saline | Mild Salinization | Moderate Salinization | Severe Salinization | Saline Soil |
|---|---|---|---|---|---|
| Proportion of inversion result | 0 | 3.00 | 43.65 | 17.13 | 36.22 |

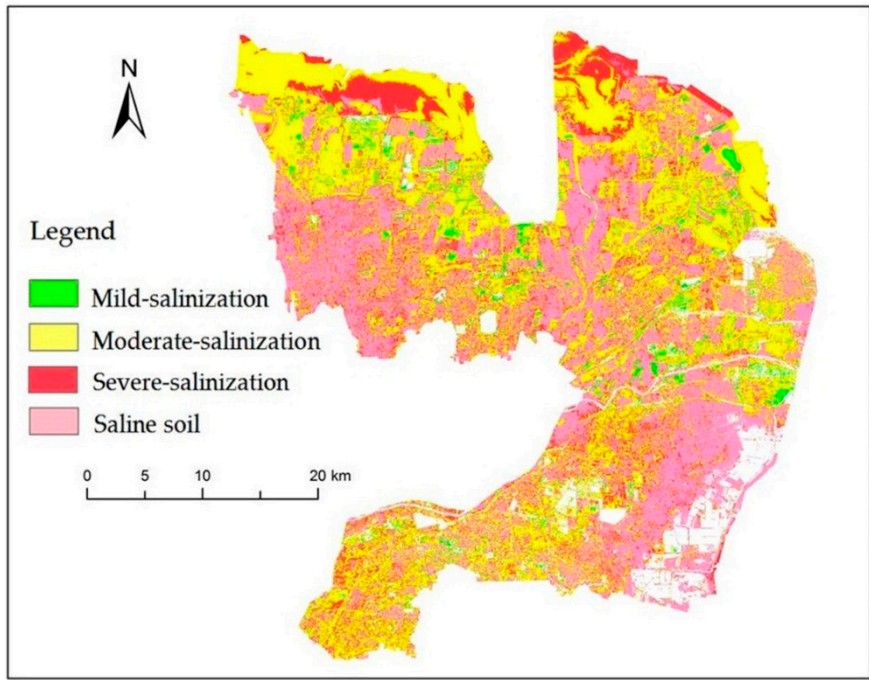

**Figure 8.** Distribution map of the inverted soil salinity in the study area in 2018.

As shown in Figure 8, salinized soil is widely distributed throughout the coastal area of the YRD and concentrated along the eastern coastline of Kenli District and the eastern and southwestern parts of Hekou District. Severely salinized soil accounts for a relatively small proportion and is concentrated in the northern part of Hekou District. In contrast, moderately salinized soil, which accounts for the largest proportion, is distributed in the middle and southwestern parts of Kenli District and the eastern and northern parts of Hekou District. Furthermore, mildly salinized soil accounts for the smallest proportion and is mainly distributed along the Yellow River and the ancient Yellow River channel, where irrigation water is abundant. Finally, non-salinized soil is not present within the study area.

In summary, the inversion results are consistent with the actual situation of the study area in 2018, demonstrating that the model is suitable for the inversion of the springtime soil salinity in the coastal area of the YRD and could play an important role in the remote sensing monitoring of soil salinity throughout the YRD.

## 4. Discussion

1.  To reduce the influence of vegetation on the reflectivity of the soil surface, this study first selected the spring season, when the vegetation coverage is low, as the research period. Second, the centers of areas with relatively uniform vegetation coverage were selected as sampling points so that vegetation factors could be taken into account during the remote sensing inversion. Furthermore, the survey found that the vegetation in the study area differed significantly under different salinization levels; thus, the vegetation cover could also be used as an indirect indicator in soil salinity monitoring [44–46].

The crops in the study area grew steadily during the survey period; thus, the time span did not affect the experiment. Moreover, the results verify the universality of the model, which is applicable for the inversion of soil salinity in spring.

2.  The maximum impact on crops is from the salt content in the surface layer of soil (0~10 cm), but the "remote sensing" method can observe only the surfaces of objects, and the "surface layer" and "surface" cannot form a completely consistent relationship. Therefore, by analyzing the relationship between the soil salinity of the surface layer and various spectral information reflected from the soil surface (the information is more or less directly or indirectly related to the soil salinity), this paper aims to construct an inversion model to estimate the salinity of the soil surface layer. In the selection of remote sensing sources and the screening of sensitive bands and spectral parameters, the interference of irrelevant information is removed to the maximum extent to improve the precision of the inversion model. However, further improvement of the precision will remain a focus of future research.

3.  The correlation of the soil salinity with UAV imagery is generally higher than that with Sentinel-2A imagery, indicating that a high spatial resolution has a significant effect on the recognition of spectral soil salinity information. The spatial resolution refers to the size of the ground area represented by a pixel. The higher the spatial resolution is, the smaller is the ground area represented by a single pixel. Due to the large difference in soil salinity in the study area, the smaller the pixel is and closer the location is to the ground sampling point, the more accurate will be the information obtained [47,48]. The centimeter resolution of the Sequoia multispectral imagery (5 cm resolution, in which 40,000 UAV pixels correspond to one satellite pixel) can effectively reduce the influence of mixed pixels and improve the purity of spectral soil salinity information. Consequently, the UAV-based soil salinity model has strong universality and high stability.

4.  Vegetation indexes, ratio indexes and normalized difference indexes were also employed in this study, but the results were not satisfactory. It is believed that with the low vegetation coverage in the study area during springtime, the significant difference and ratio relationship between red light and near-infrared light no longer exists, thus affecting the effect of the two matching methods of the ratio and normalized difference. Similarly, due to the low vegetation coverage, the differences in the vegetation indexes are not as obvious as those in the soil salinity in the study area, so the spectral characteristics of soil salinity cannot be expressed. In addition, previous studies have explored the relationship between vegetation indexes and soil salinity and showed that the R values are low [49–51]. Further research has shown that vegetation indexes are not efficient in soil salinity monitoring and has recommended the construction of spectral parameters via band combinations [52,53]. Therefore, different spectral parameters should be selected for different environments [54,55]. In this paper, addition and multiplication methods, which can increase the effective spectral information in a single band, mutually increasing and improving the expression ability of soil spectral characteristics, are more suitable for the YRD in spring.

5.  In contrast with previous studies, this paper integrates multiple sensors, multiple modeling methods, and multiscale mapping and achieves high-precision, multiscale, high-spatial and high-temporal-resolution inversion of soil salinity in the study area [56–58]. However, there are still three aspects that need to be further studied. First, the precision needs to be further improved, and the contributions of different data sources and nonlinear modeling methods to the precision will continue to be explored [59,60]. Additionally, the driving factors and formation mechanism of soil salinization must be elucidated to prevent this problem [61,62]. Furthermore, the universality of the time and space of monitoring should be expanded [63,64].

## 5. Conclusions

This paper proposes a method for inverting the salinity of coastal saline soil. A model was constructed and the multiscale inversion of soil salinity was realized. The following conclusions can be drawn.

1. The sensitive to soil salinity are the Bg (green), Br (red), Breg (red-edge) and Bnir (near-infrared bands). The spectral parameters are mainly Bg × Breg and Bg + Breg, among others, with Bg exhibiting the strongest response to soil salinity.

2. In this paper, the best inversion model of coastal saline soil salinity is as follows: $Y = 27.62 - 164.215 \times Bg - 12.307 \times Br - 29.673 \times Bnir - 5.19 \times Breg + 316.529 \times Bg \times Bnir - 134.333 \times Bg \times Bnir \times Breg + 603.149 \times Bg \times Br \times Breg - 1509.075 \times Bg \times Br \times Bnir \times Breg + 415.695 \times Bg^3$. The modeling precision of this model is 0.743, and the verification precision is 0.809. The model has high significance, strong predictive ability and good universality.

3. The UAV-based model cannot be applied directly to the Sentinel-2A images to achieve scaled-up inversion. Therefore, the normalization correction of reflectivity is a necessary process.

4. This paper presents an inversion of the soil salinity at different scales, thereby confirming the feasibility, reliability and stability of the proposed method of performing a scaled-up inversion of the large-scale soil salinity in satellite images with reflectance correction.

This paper presents an effective method to map the soil salinity in the coastal area of the YRD by integrating satellite, UAV and ground-measured data. A highly universal inversion model of the springtime soil salinity is constructed accordingly and a scaled-up inversion method for application to satellite imagery is proposed, which can perform accurate soil salinity inversions for coastal saline soils at different scales. This method shows potential to be used in the management and utilization of salinized land sources.

In the future, the following three aspects will be focused on: the improvement of model precision, the discussion of the driving factors and formation mechanism of soil salinization and the improvement of spatial-temporal universality of monitoring.

**Author Contributions:** Conceptualization, G.Z. and S.Z.; Investigation, S.Z.; Methodology, G.Z. and S.Z.; Validation, G.Z.; Writing—original draft, G.Z. and S.Z.; Writing—review & editing, S.Z.

**Funding:** This work was financially supported by the National Natural Science Foundation of China (41877003), the Chinese Science and Technology Projects (2015BAD23B0202 and 2013BAD05B0605), and the Funds of Shandong "Double Tops" Program (SYL2017XTTD02). We also thank American Journal Experts for the language editing.

**Acknowledgments:** Thanks to American Journal Experts for the language editing. Thanks to Baowei, Su, Xiaona, Chen, Kun, Lang, Xue, Xi for Investigation.

**Conflicts of Interest:** There are no conflict of interest exits in the submission of this manuscript, and manuscript is approved by all authors for publication. I would like to declare on behalf of my co-authors that the work described was original research that has not been published previously, and not under consideration for publication elsewhere, in whole or in part. All the authors listed have approved the manuscript that is enclosed.

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
