# Peer review of "A Harmonious Satellite-Unmanned Aerial Vehicle-Ground Measurement Inversion Method for Monitoring Salinity in Coastal Saline Soil"

_remotesensing, doi:10.3390/rs11141700_

Round 1

Reviewer 1 Report

Please, see the attached fiel

Author Response

Dear Editor:

Thank you for handling our manuscript entitled “A Harmonious Satellite-Unmanned Aerial Vehicle (UAV)-Ground Measurement Inversion Method for Monitoring Salinity in Coastal Saline Soil” (No: remotesensing-521666). We appreciate the comments from the reviewers, which helped to improve the manuscript significantly. In the following section, we explain in detail how we responded to each of the comments.

Response to Reviewer 1 Comments

Point 1: The topic of this paper focuses on the soil salinity mapping and monitoring using UAV, Sentinel-MSI and field soil sampling. The paper subject is appropriate and conform to the Remote Sensing journal objectives, unfortunately the paper show several weaknesses in the different steps of its presentation, in same sections it is difficult to follow or to understand, and English language request major revision. Consequently, the paper needs major revision and improvement. 

Response 1: We are very grateful for your recognition of this research. We have seriously modified the deficiencies of the article, and have revised the English language by American Journal Experts. We are very grateful to the editors and reviewers for your hard work.

Point 2:Lines 29 – 32: this sentence is not so clear.

Response 2:Thanks for the reviewer's reminding, and we added the word that was missing in the editing process.

Page 1 Line 27-29: This study integrated the advantages of satellite, UAV and ground methods and then proposed a method for the inversion of the salinity of coastal saline soils at different scales

Point 3:L 49: remote sensing is a science and only a technique! 

Response 3:Thanks for the reviewer's reminding,and we have modified the wrong wording.

Page 2 Line 47-48: satellite remote sensing has gradually replaced traditional soil salinity monitoring methods, which are inefficient and expensive.

Point 4:L 82 – 83: this sentence is not so clear.

Response 4:Thanks for the reviewer's reminding, and we adjusted the order of words and added explanations.

Page 2 Line 82-84: When building the satellite-based model, a ground sampling point data corresponds to a pixel of satellite imagery. The method for building the satellite-based model is the same as that used for the UAV-based model, as shown in section 2.2.

Point 5:L 126: if the accuracy of your GPS is around 3-25 mm means it is a differential GPS (DGPS). Please, clarification is requested.

Response 5:Thanks for the reviewer's reminding.We did use DGPS. According to the comment of reviewer, we have clarification it.

Page 3 Line 120-122: The coordinates (longitude and latitude) of the sampling points were measured with a Trimble GEO 7X centimeter handheld differential GPS (DGPS) with an accuracy of approximately 3~25 mm.

Point 6:L 126: Since you have the measurements of electrical conductivity (Ec) you MUST provide scatter-plots between Ec and reflectances in each spectral band (UAV and Sentinel sensors) considering your 140 sampling points.

Response 6: Scatter diagrams have been added as suggested by the reviewer. Page 6-7, Figure 2.

Point 7:L 127: theoretically your 69 verification points MUST be in areas with different salinity degrees because your research paper is about salt-affected soil modeling and not about geomorphology and land use classes.

Response 7:Thanks for the reviewer's reminding, andwe have supplemented it.

Page 3 Line 129-Page 4 Line 131: The measured soil salinity values were significantly different, with different salinity degrees (as high as 9.52 g/kg, which indicates saline soil, and as low as 1.34 g/kg, which indicates nonsalinization).

What needs to be explained is that verification points were randomly selected according to land use type,because the verification set should not only verify the verification accuracy of inversion model, but also verify the universality of the model. Different land use type can prove that the model can be applied not only to farmland, but also to other fields.At the same time, soil salinity also varies with different land use types(for example, soil salinity in agricultural land is relatively low, while that in bare land is relatively high).

Point8: Lines 139 to 143: the UAV radiometric and spectral calibration is not clear and need more clarification. In addition, Sentinel-MSI data are coded in 12 bit while the UAV data are in 8 bit and these differences have a significant impact.

Response 8: According to the reviewer’s suggestion, we have further clarified the UAV radiometric and spectral calibration.

Page 4 Line 139-144: During the process of taking photographs, the sunshine sensor equipped with the Sequoia could correct the illumination difference to calibrate the intrinsic radiation. By taking the whiteboard images by the Sequoia camera and loading the whiteboard images and whiteboard parameters corresponding to each bandin the Pix4D software before image mosaicking, extrinsic radiation calibration and spectral calibration were achieved.

As for the coding of Sentinel-MSI and UAV, although the coding of them are different, we have converted the DN value into the reflectivity value (between 0 and 1) in the image pre-processing, so the two are comparable and have no impact.

Point9: L 200: which model are you talking about?

Response 9: Thanks for the reviewer's reminding,and we have corrected.

Page 5 Line 209-210: The validation of the UAV-based best soil salinity inversion model obtained from section 2.2 was mainly reflected in the following two aspects.

Point10: Section 3.1.1.:I don’t believe in these results, please you MUST provide scatter-plots between Ec and reflectance in each spectral band (UAV and Sentinel sensors) considering your 140 sampling points.

Response 10: Scatter diagrams have been added as suggested by the reviewer. Page 6-7, Figure 2.

Point11: Indeed, the literature demonstrated that in the VNIR we have severe confusion between soil salinity classes and soil optical properties (color, brightness, texture, moisture, mineralogy, organic matter, etc.). As well, the correlation between the Ec and the reflectances in the VNIR bands are always less than 5%.Please, see the following references…

Response 11: We strongly agree with the reviewer's view that there will be confusion between soil salinity classes and soil optical properties, and we have carefully read the references recommended by the reviewer (except the totally unsearched references 2, 7 and 8), thus leading to the following thoughts:

Firstly, most of the research areas selected in the literature are arid or extremely arid inland areas [1,5,9] and most of them have tropical climate [1,4,6,9], which makes the soil type, geological characteristics, climate conditions and causes of salinization different from this study, and the difference of test results is inevitable.Even though the sampling time is similar to that of this study, for example, in reference 1, the sampling date is from March to April, but the temperature of the two places is different.In reference 1, the highest temperature in Bahrain has reached about 25°C, while YRD in this research is only about 10°C. The response of crops to soil salinity is different under different temperatures, which leads to the phenomenon of different spectral reflectivity and salt response.

Secondly, except for reference 1, most of the other references are from 15 years ago. Although they represent the cutting-edge scientific research achievements at that time, the limitations mentioned in them have been widely paid attention to and avoided by various means.For example, it is mentioned in reference 2 that “Constraints on the use of remote sensing data for mapping salt-affected area are shown related to the spectral behaviour for salt type, spatial distribution of salt on the terrain surface, temporal changes on salinity, inference of vegetation, and spectral confusions with other terrain surfaces.”In this study, the soil in the whole study area is coastal saline soil, so the salt type is the same.The whole research area is a plain with flat terrain, and the terrain factors can also be ignored.The interval between the two sampling was only one year, and both samples were in March. At the same time, the vegetation coverage was low and salt precipitates to the surface of the soil, which was the best time for soil surface salinity monitoring, which was consistent with “salt identification is easiest at the end of the dry season.”and “with low moisture content, salt were sensitive in the visible region of the spectrum.” from reference 1.The high spatial resolution airborne multispectral camera can effectively reduce the influence of spectral confusion.

Finally, thanks to the efforts of scientists in the research and development of remote sensing data source equipment, the early research can only be based on Landsat, SPOT [6]and other data sources, which have long cycle, small number of wavelengths, and low spatial resolution.Sentinel data overcomes these limitations and provides us with better scientific data sources.

I have benefited a lot from the references provided by the reviewers. However, in order to enhance your trust in the research results, I have also searched several literatures with relevance to this research in limited time [A, B, C, D, E], which all reflect the high correlation between Ec and reflectance in the VNIR.

Reference B:The correlation between Ec and each band of digital camera is shown in the following table. (The table will be dispalyed in the WORD)

Reference C:The relationship between the Ecof 0~250px surface soil and Landsat bands is shown in the following table. (The table will be dispalyed in the WORD)

Digital cameras with lower spatial resolution than Sequoia and Landsat satellites with lower spatial resolution than Sentinel were able to get such high correlations with Ec,so the correlation between Sequoia and Sentinel withEc in this study is practical [E].

A.  Bai, L.; Wang, C.Z.; Zang, S.Y.; Wu, C.S.; Luo, J.M.; Wu, Y.X. Mapping Soil Alkalinity and Salinity in Northern Songnen Plain, China with the HJ-1 Hyperspectral Imager Data and Partial Least Squares Regression. Sensors 2018, 18, doi:10.3390/s18113855.

B.  Celleri, C.; Zapperi, G.; Trilla, G.G.; Pratolongo, P. Assessing the capability of broadband indices derived from Landsat 8 Operational Land Imager to monitor above ground biomass and salinity in semiarid saline environments of the Bahia Blanca Estuary, Argentina. International Journal of Remote Sensing 2019, 40, 4817-4838, doi:10.1080/01431161.2019.1574992.

C.  Jiang, H.N.; Shu, H. Optical remote-sensing data based research on detecting soil salinity at different depth in an arid-area oasis, Xinjiang, China. Earth Sci. Inform. 2019, 12, 43-56, doi:10.1007/s12145-018-0358-2.

D.  Xu, L.; Zheng, C.L.; Wang, Z.C.; Nyongesah, M.J. A digital camera as an alternative tool for estimating soil salinity and soil surface roughness.Geoderma 2019, 341, 68-75, doi:10.1016/j.geoderma.2019.01.028.

E.   Yu, H.; Liu, M.Y.; Du, B.J.; Wang, Z.M.; Hu, L.J.; Zhang, B. Mapping Soil Salinity/Sodicity by using Landsat OLI Imagery and PLSR Algorithm over Semiarid West Jilin Province, China. Sensors 2018, 18, doi:10.3390/s18041048.

Point12: Lines 237-241: why the correlation between Sentinel data and UAV is weak?

Response 12: Sorry, I didn't understand you exactly. The paper mentionedthat “The correlation between the reflectivity of Sentinel-2A imagery and the soil salinity is weaker than that of the UAV imagery.” There is no mention of the correlation between Sentinel and UAV. Did my language misunderstand you?

Point13: Lines 237-241: which model are you talking about? 

Response 13: Thanks for the reviewer's reminding,and wehavefurther explained the model mentioned.

Page 6 Line 248-249: Therefore, it is not difficult to infer that the soil salinity inversion model based on UAV imagery is more accurate than the model based on Sentinel-2A imagery.

Point14: Section 3.1.2. (Table 4): Please, you MUST provide scatter-plots for these correlations. According to my 20 years’ experience working on this topic I don’t believe in these correlation values!!! 

Response 14: Scatter diagrams have been added as suggested by the reviewer. Page 8, Figure 3.

Point15: Section 3.3.1.: this section is very poor and the analysis is very superficial. Please, the curves in Figure 2a are different in amplitude with 10%?

Response 15: I'm afraid I didn't quite understand your question, but we tried to explain it.

Page 8, Line 290-292: The amplitude of the UAV curve is 0.038962, the amplitude of the Sentinel-2A curve is 0.058313, and the amplitude difference between the two curves is 0.013693.

Point16:What is the significance ofFigure 2b results using only 4 points (approximately) for statistical correlation? However, in your soil survey you have 140 points!

Response 16: Thanks for the reviewer's reminding, we have supplemented the production of Figure 2.

Page 5, Line 193-198: To determine whether it is feasible to normalize the reflectivity of Sentinel-2A images based on UAV images, the average reflectivity of 140 sampling points in each band of the UAV and Sentinel-2A images was calculated, and the variation trends of the two images were compared. Furthermore, scatter plots of the average reflectivity of the corresponding bands of the UAV and the Sentinel-2A images were generated to prove the correlation between the reflectivities of the two images.

Point17: Section 3.3.2.: very poor analysis. Please, what is your argumentation regarding the low correlation in short wavelengths? More and deep interpretations are requested.

Response 17: According to the reviewer’s suggestion, we have added some interpretations for this part.

Page 9, Line 307-312: This phenomenon may be affected by the following three factors: first, the difference between the Sentinel-2A and UAV images in the center wavelength of the Bg band is 10nm, which is the largest difference among all the bands; moreover, the gap between Sentinel-2A and UAV in the Bg band is the largest of all the bands (Figure 4 (a)); and it was hypothesized that the spatial resolution of the Sentinel-2A image is low, which is affected by the spectral confusion caused by the canopy of only a few crops, and the Bg and Br bands were the most affected.

Point18:Table 7: the case-1 results using only the VNIR bands provide the evidence that this region of wavelengths is not appropriate for soil salinity classes’ discrimination. Indeed, your R2 is only 0.45 for modeling and 0.30 for validation.

Moreover, I don’t believe that if you play with bands number in your equations using different power degrees you will be able to improve the accuracy of your results. 

Response 18: This should be a misunderstanding, and you may not have noticed that the inversion model of case (2) and case (3) is based on the UAV imagery, and the inversion model of case (1) is based on the Sentinel-2A imagery. The differenceof data sources lead to different modelingprecision, which leads to different validation accuracy when applied to Sentinel-2A images before and after reflectivity normalization.

Reviewer 2 Report

I have no major concern about this paper, but some improvements are necessary before publication. Some suggestions are the following:

1)      The abstract is too long. Be more succinct.

2)      Please, clear state the contributions of the paper in the introduction.

3)      The authors mention some related works in the introduction, but the paper lacks a serious discussion about related works.

4)      Please, refrain from using the first person, “we, our, us”, please prefer using third person or passive voice instead (direct speech is always better, you should always avoid first person!).

5)      The conclusion should have one final paragraph in which the authors present directions for future work.

Author Response

Dear Editor:

Thank you for handling our manuscript entitled “A Harmonious Satellite-Unmanned Aerial Vehicle (UAV)-Ground Measurement Inversion Method for Monitoring Salinity in Coastal Saline Soil” (No.: remotesensing-521666). We appreciate the comments from the reviewers, which helped to improve the manuscript significantly. In the following section, we explain in detail how we responded to each of the comments.

Response to Reviewer 2 Comments

Point 1:I have no major concern about this paper, but some improvements are necessary before publication. Some suggestions are the following

Response 1: We are very grateful for your recognition of this research. We will seriously modify the deficiencies of the article. We are very grateful to the editors and reviewers for your hard work.

Point 2:The abstract is too long. Be more succinct.

Response 2:According to the comments of reviewers, we have simplified the abstract.

Point 3: Please, clear state the contributions of the paper in the introduction.

Response 3: According to the comments of reviewers, wehave added the contribution of this paper in the introduction.

Page 2 Line 76-78: The advantages of ground-measured, UAV and satellite methods were fully harmonized in this study. In addition, the scale, precision and spatial-temporal resolutions of soil salinity inversions were improved.

Point 4: The authors mention some related works in the introduction, but the paper lacks a serious discussion about related works.

Response 4: According to the comments of reviewers, we supplemented the discussion about related work.

Page 2 Line 65-68: However, most data sources are the synergies of radar and optical data and the combination of various satellite remote sensing data. Most of the research objects focus on vegetation canopy parameters. Accordingly, the harmonious use of satellite, UAV and ground data should be rare and feasible and is thus worthy of further research.

Point 5: Please, refrain from using the first person, “we, our, us”, please prefer using third person or passive voice instead (direct speech is always better, you should always avoid first person!).

Response 5: Thanks for the reviewer's reminding, we have modified all the first person.

Point 6: The conclusion should have one final paragraph in which the authors present directions for future work.

Response 6: As suggested by the reviewer, we have added one final paragraph in the conclusion to explain the future work directions.

Page 14 Line 463-465: In the future, the following three aspects will be focused on: the improvement of model precision, the discussion of the driving factors and formation mechanism of soil salinization, and the improvement of spatial-temporal universality of monitoring.

This manuscript is a resubmission of an earlier submission. The following is a list of the peer review reports and author responses from that submission.

Round 1

Reviewer 1 Report

how was the average soil sanlity collected? The authors stated the salinity was measured for the first 10cm (z direction) of soil, but how about in the x and y direction? was it just one measurement at a specific pin point location? was is measureed multiple times in a vicinity? In addition, the instrument measures the first 10cm of the ground, however, the UAV sensor only "senses" the surface, not the first 10cm. Can this introduce any error in the analysis? 

Can the authors provide manfacture / model information about the EC110 portable salinity meter used in the study?

what was the accuracy / potential error of the GPS measurements?

how was the Sequoia sensor calibrated (intrinsic and extrinsic)? how was the UAV imagery processed and corrected? How was the data converted to surface reflectance? What are the processing steps? How were the steps excuted? Was the result validated using any ground truth?

table 1, in addition to central wavelenth, FWHM should be provided

in the manuscript, it seems that the authors corrects Sentinel-2A reflectance to match UAV reflectance, what's the reason to do that? why not the other way around? i.e. correcting UAV imagery to match surface reflectance derived from Sentinel data? which are acquired by a more sophiscated sensor and corrected with a more robust algorithm?

why was the reflectance correcion was done only by applying a ratio, not a ratio and an offset (i.e., in  the concept of gain and offset or slope and intercept)

how was Figure 2(a) and (b) derived? did the authors take one pixel from each band? what are these points?

Can the authors provide the scatter plot of UAV reflectance vs "corrected" Sentinel for each band? Fig2b shows a strong correlation but far away from the 1:1 line, is this before correction? how about after correction? 

line 175: "coefficient of determination"

line 237: why "obviously"??

line 237 to 242: how does the significantly different spatial resolution (pixel size) of the two imagery (4-5cm vs 10m) affect the correlation? if at all?

Table 3: why these combination? did the authors try anything in a simple ratio index and normailized difference index format?

line 373 to 377: how was this affected by field sampling strategy? for instance, if the field samples were collected uniformily withthin a 10mx10m area, would that increase the correlation between field vs sentinel?

Author Response

Dear Editor:

Thank you for handling our manuscript entitled “Satellite-Unmanned Aerial Vehicle (UAV)-Ground Harmonious Inversion Method of Monitoring Salinity in Coastal Saline Soil” (No.: remotesensing-477200). We appreciate the comments from the reviewers, which helped to improve the manuscript significantly. In the following section, we explain in detail how we responded to each of the comments.

Response to Reviewer 1 Comments

Point 1: How was the average soil salinity collected? The authors stated the salinity was measured for the first 250px (z direction) of soil, but how about in the x and y direction? Was it just one measurement at a specific pin point location? Was it measured multiple times in vicinity? 

Response 1: Thanks for the reviewer's reminding, and we have supplemented the collection method of salt salinity.

Page 3 Line 125-128: An EC110 portable salinity meter equipped with 2225FS T series probe (in which the temperature correction for the electrical conductivity had already been completed) (Spectrum Technologies, Inc., USA) was used to make multiple measurements at and near the sampling points, with a range of no more than 5 cmĂ—5 cm.

Point 2: In addition, the instrument measures the first 250px of the ground; however, the UAV sensor only "senses" the surface, not the first 250px. Can this introduce any error in the analysis?

Response 2: Thanks for the reviewer's reminding, and we explained this problem in Discussion.

Page 12 Line 384-389: The maximum impact on crops is the salt content in the surface layer of soil (0 ~10 cm), but the "remote sensing" method can only observe the surface of objects, and the two cannot form a completely corresponding relationship. Therefore, by analysing the relationship between the soil salinity of the surface layer and various spectral information reflected from the soil surface (information is more or less directly or indirectly related to the soil salinity), this paper aims to construct an inversion model to estimate the salinity of soil surface layer.

In other words, the purpose of this paper is to build an inversion model of soil salinity through the relationship between surface (not surface layer) spectral information and soil salinity of surface layer.

Point 3: Can the authors provide manufacture / model information about the EC110 portable salinity meter used in the study?

Response 3: Thanks for the reviewer's reminding, and the information about the EC110 portable salinity meter has been supplemented.

Page 3 Line 125-128: An EC110 portable salinity meter equipped with 2225FS T series probe (in which the temperature correction for the electrical conductivity had already been completed) (Spectrum Technologies, Inc., USA) was used to make multiple measurements at and near the sampling points, with a range of no more than 5 cmĂ—5 cm.

Point 4: What was the accuracy / potential error of the GPS measurements?

Response 4: Thanks for the reviewer's reminding, and the accuracy error about the GPS has been supplemented.

Page 4 Line 132-133: the accuracy of the GPS measurements is about 3~25mm.

Point 5: How was the Sequoia sensor calibrated (intrinsic and extrinsic)? How was the UAV imagery processed and corrected? How was the data converted to surface reflectance? What are the processing steps? How were the steps executed? Was the result validated using any ground truth?

Response 5: Thanks for the reviewer's reminding, and we supplemented the information about the processing of Sequoia sensor in 2.2.2.

Page 4 Line 145-151: During the photo-taking process, the sunshine sensor equipped with the Sequoia can correct the illumination difference to calibrate the intrinsic radiation. Pix4D software was used to preprocess the UAV imagery, which included operations such as mosaicking, converting the data to surface reflectance and extrinsic radiation calibration. Extrinsic radiation calibration was achieved by loading the whiteboard image taken by the Sequoia camera and inputting the parameter of the whiteboard into the Pix4D software.

Point 6: Table 1, in addition to central wavelength, FWHM should be provided.

Response 6: Thanks for the reviewer's reminding, and the FWHM of Sequoia sensor has been provided.

Point 7: In the manuscript, it seems that the authors correct Sentinel-2A reflectance to match UAV reflectance, what's the reason to do that? Why not the other way around? i.e. correcting UAV imagery to match surface reflectance derived from Sentinel data? Which are acquired by a more sophisticated sensor and corrected with a more robust algorithm?

Response 7: We have described the generation of this idea in the Introduction, and we hope the following explanation will give you a better understanding.

The goal of this article is to fully utilize the individual advantages of ground, UAV and satellite methods and achieve comprehensive improvements in the scale, accuracy, and spatial-temporal resolutions of soil salinity inversions. Therefore, we choose the model of "highest precision" to apply to the "largest-scale" image, which model has the highest precision is shown by the results, not by the sophistication of the sensor. Due to the data source of "highest precision" model is different from the "largest-scale" image, the data source of "highest precision" model is UAV and of the "largest-scale" image is Sentinel-2A, and the validation results showed that the model cannot be directly applied to the image, so normalization of Sentinel-2A imagery to UAV imagery is required.

Point 8: Why was the reflectance correction was done only by applying a ratio, not a ratio and an offset (i.e., in the concept of gain and offset or slope and intercept)

Response 8: Thanks for the reviewer's reminding. I think my wrong words may have misled you. The "correction" in this paper is not the well-known "radiation correction" of satellite images, but the normalization of two different data sources, so as to make the reflectivity of Sentinel-2A images consistent with the UAV images. I am deeply sorry for the loose words, and I have revised them in the full text.

Point 9: How was Figure 2(a) and (b) derived? Did the authors take one pixel from each band? What are these points?

Response 9: Thanks for the reviewer's reminding, and we made a more detailed supplement in the Second 2.

Page 2 Line 89-94: The data sources in this paper include satellite, UAV and ground measured data. When building the UAV-based model, a ground sampling data point corresponds to a pixel of UAV imagery. When building the satellite-based model, a ground sampling data point corresponds to a pixel of satellite imagery. When calculating the normalization coefficient of reflectivity of the satellite images, there are approximately 1600 UAV image pixels corresponding to a satellite image pixel in the same area.

Page 5 Line 196-197: The average reflectivity of each band of the 1600 pixels of UAV images and the corresponding pixel of Sentinel-2A imagery were obtained, and the relationship between them was analyzed.  

Point 10: Can the authors provide the scatter plot of UAV reflectance VS "corrected" Sentinel for each band? Fig2b shows a strong correlation but far away from the 1:1 line, is this before correction? How about after correction?

Response 10: Thanks for the reviewer's reminding. Fig 2 (b) is before correction (normalization), and the Figure 3 we added is after correction (normalization).

Point 11: line 175: "coefficient of determination"

Response 11: Thanks for the reviewer's reminding, and the wrong words have been corrected.

Page 6 Line 189: "decision coefficient" has been changed to "coefficient of determination".

Point 12: line 237: why "obviously"?

Response 12: Thanks for the reviewer's reminding, and the inappropriate words have been deleted.

Point 13: line 237 to 242: how does the significantly different spatial resolution (pixel size) of the two imagery (4-125px VS 10m) affect the correlation? If at all?

Response 13: This point has been discussed in the Discussion, but it may not be clearly expressed. According to the suggestion of reviewer, we have enriched it.

Page 12 Line 393-403: The correlation of the soil salinity with UAV imagery is generally higher than that with Sentinel-2A imagery, indicating that a high spatial resolution has a significant effect on the recognition of spectral soil salinity information. The spatial resolution refers to the size of the ground area represented by a pixel. The higher the spatial resolution is, the smaller the ground area represented by a single pixel. Due to the large difference in soil salinity in the core test area, the smaller the pixel is and the closer the location is to the ground sampling point, the more accurate the obtained information will be. The centimeter resolution of the Sequoia multispectral imagery (5 cm resolution, in which 40,000 UAV pixels correspond to one satellite pixel) can effectively reduce the influence of mixed pixels and improve the purity of spectral soil salinity information. Consequently, the UAV-based soil salinity model has strong universality and high stability.

Point 14: Table 3: why these combinations? Did the authors try anything in a simple ratio index and normalized difference index format?

Response 14: This point was also briefly discussed in the Discussion. According to the reviewer’s suggestion, this part was been enriched.

Page 13 Line 404-417: Vegetation indexes, ratio indexes and normalized difference indexes were also employed in this study, but the results were not satisfactory. It is believed that with the low vegetation coverage in the study area during springtime, the significant difference and ratio relationship between red light and near-infrared light no longer exists, thus affecting the effect of the two matching methods of ratio and normalized difference. Similarly, due to the low vegetation coverage, the difference in vegetation index is not as obvious as that of soil salinity in the study area, so the spectral characteristics of soil salinity cannot be expressed. In addition, previous studies have explored the relationship between vegetation indexes and soil salinity and showed the R value is approximately -0.4. However, further research has shown that vegetation indexes are not efficient in soil salinity monitoring and recommended the construction of spectral parameters via band combinations. Therefore, different spectral parameters should be selected for different environments. Addition and multiplication methods, which can increase the effective spectral information of a single band mutually gain and improve the expression ability of soil spectral characteristics, are more suitable for the YRD in spring.

Point 15: How was this affected by field sampling strategy? For instance, if the field samples were collected uniformly within a 10mx10m area, would that increase the correlation between field VS sentinel?

Response 15: This question is the key questions we considered when designing experiments. In this paper, 160 observation areas of 20mĂ—20m were established (added in Page 3 Line 124), and their central points were taken as sampling points (2.2.1) which means that the field samples were collected uniformly. As for area, reducing the sampling area may increase the correlation between field data and remote sensing data. However, for red edge band of Sentinel-2A with 20m resolution, collect the field samples uniformly within a 10mx10m area were meaningless. Therefore, in consideration of data integrity, the sampling area of this study is 20mĂ—20m.

Reviewer 2 Report

The manuscript begins nicely and clearly, but later in the Materials and Methods and later becomes not enough described. Discussion is weak.  Conclusions are inaccurate and potentially incorrect (must be rewritten)! English is good but style varies from technicaly clear/perfect to somewhat superficial.

I expect explanation/description of applied Remote Sensing inversion method, which is the core point respecting the title of the paper.

I was expecting more than 30 lines of discussion and more comparison with recent works on soil salinity monitoring.

Author Response

Dear Editor:

Thank you for handling our manuscript entitled “Satellite-Unmanned Aerial Vehicle (UAV)-Ground Harmonious Inversion Method of Monitoring Salinity in Coastal Saline Soil” (No.: remotesensing-477200). We appreciate the comments from the reviewers, which helped to improve the manuscript significantly. In the following section, we explain in detail how we responded to each of the comments.

Response to Reviewer 1 Comments

Point 1: The manuscript begins nicely and clearly, but later in the Materials and Methods and later becomes not enough described. Discussion is weak. Conclusions are inaccurate and potentially incorrect (must be rewritten)! English is good but style varies from technically clear/perfect to somewhat superficial.

Response 1: We are very grateful for your recognition of this research. We will seriously modify the deficiencies of the article. We are very grateful to the editors and reviewers for your hard work.

As suggested by the reviewer, we have supplemented the Materials and Methods to make it clearer, enrich the Discussion, rewritten the Conclusions to make it accurate and correct, and made extensive revisions to the English.

Point 2: I expect explanation / description of applied Remote Sensing inversion method, which is the core point respecting the title of the paper.

Response 2: We agreed with the reviewer’s comment, and enhance the explanation and description of applied Remote Sensing inversion method in Materials and Methods and Discussion.

Point 3: I was expecting more than 30 lines of discussion and more comparison with recent works on soil salinity monitoring.

Response 3: As suggested by the reviewer, we supplement more comparison with recent works on soil salinity monitoring in Discussion.

Page 13 Line 418-425: In contrast with previous studies, this paper integrates multi-sensors, multi-modeling methods and multi-scale mapping and achieved high precision, high spatial and temporal resolution, and multi-scale inversion of soil salinity in the research area [61-63]. However, there are still three aspects that need to be further studied. First, the precision needs to be further improved, and contributions of different data sources and nonlinear modeling methods to the precision will continue to be explored [64,65]. At the same time, the driving factors and formation mechanism of soil salinization must be elucidated to prevent this problem [66,67]. Furthermore, the universality of time and space of monitoring should be improved [68,69].

Point 4: Generally the conclusions are imprecisely written! Improve / rewrite, please.

Response 4: As suggested by the reviewer, we rewritten and improved the conclusions generally.

Point 5: Line 394-395, this is not info to “Conclusion”.

Response 5: As suggested by the reviewer, this part has been deleted.

Point 6: Line 399-401, I am not sure from what this conclusion means and from where it comes? Maybe, English should be improved!?!

Response 6: As suggested by the reviewer, we improved the English, and the conclusion is from 3.1.1, 3.1.2

Page 14 Line 463-466: The sensitive bands of soil salinity are the Bg (green), Br (red), Breg (red-edge) and Bnir (near-infrared) bands. The spectral parameters are mainly BgĂ—Breg and Bg+Breg, among others, where Bg exhibits the strongest response to soil salinity.

Point 7: Line 404, delete.

Response 7: As suggested by the reviewer, the extra “.” has been deleted.

Point 8: Line 406, I wouldn’t say “ALL” (btw, it would be interesting to see somewhere in the text what kind of soil dominates in your research area).

Response 8: As suggested by the reviewer, the incorrect “ALL” has been deleted.

Point 9: NOT “springtime soils” (the spring is 21 March - 21 June! March 5-10, 2018 was still wintertime) please, carefully, reformulate your conclusion.

Response 9: Thanks for the reviewer's reminding. The division method you mentioned is "astronomical division", which shows great disadvantages in China due to the wide territory. As shown in the Figure (1), even in the same region, the seasons at the same time are also different. Therefore, China's meteorological departments adopt the "phenology temperature method" to divide the 4 seasons, as shown in Table (1), 5 days is a phenology. Therefore, the spring in the study area (China) is from March 1 to May 31. We hope you will accept this explanation.

(The explanatory diagrams and tables are displayed in WORD)

Point 10: Line 408-410, this should go to “result”.

Response 10: Thanks for the reviewer's reminding, and this part has been deleted.

Point 11: Line 410-411, again “springtime”.

Response 11: The same Response 9

Point 12: Line 420-421, I would say: “This method show potential to be built in as one of procedures for the management and utilization of salinized land resources.”…or something like that…

Response 12: Thanks for the reviewer's suggestion, and we have revised the statement in this sentence.

Page 16 Line 449-450: This method show potential to be built in as one of procedures for the management and utilization of salinized land sources.

Reviewer 3 Report

In paragraph 2, there is an error in writing the sequence (2.1, 2.1.1, 2.1.2..)

The table 6  to make more readable

The paper is good , for me is necessary a logical synthesis of  results and analysis .

From pargraph 2.2

Line 161 …we perfomed  (not we, but in impersonal form).From line 175 to line 180…for me  s is better to explain with the help of a table or figure What do you think?

Paragraph 3.4.2 ..the inversion and validation in the area (Kenli District..) It’ better summarize all the results witouth ripetitions, with patterns and tables and synthesizing.

In the conslusions  it is advisable to define the results from future approaches and the consequences of the study

Author Response

Dear Editor:

Thank you for handling our manuscript entitled “Satellite-Unmanned Aerial Vehicle (UAV)-Ground Harmonious Inversion Method of Monitoring Salinity in Coastal Saline Soil” (No.: remotesensing-477200). We appreciate the comments from the reviewers, which helped to improve the manuscript significantly. In the following section, we explain in detail how we responded to each of the comments.

Response to Reviewer 1 Comments

Point 1: In paragraph 2, there is an error in writing the sequence (2.1, 2.1.1, 2.1.2.). 

Response 1: Thanks for the reviewer's reminding, and the error of the sequence has been corrected.

Point 2: The table 6 to make more readable.

Response 2: As suggested by the reviewer, Page 11 Table 6 has been modified. I hope my revision made Table 6 more readable with the textual explanation.

Page 9 Line 304-309:

Table 7 presents a comparison of the precision of the following three cases: (1) the soil salinity inversion model based on Sentinel-2A imagery is applied directly to the satellite imagery; (2) the best soil salinity inversion model based on UAV imagery is applied to the Sentinel-2A images before reflectance correction is performed; and (3) the best soil salinity inversion model based on UAV imagery is applied to the Sentinel-2A images after reflectance correction is performed.

(The Table is displayed in WORD)

Point 3: The paper is good, for me is necessary a logical synthesis of results and analysis.

Response 3: We are very grateful for your recognition of this research. And we have generally enhanced the logical synthesis of results and analysis.

Point 4: Line 161 …we performed (not we, but in impersonal form).

Response 4: Thanks for the reviewer's reminding, it has been changed into impersonal form.

Page 7 line 166-168: the satellite bands that are consistent with the wavelength range of the Sequoia camera are selected.

Point 5: From line 175 to line 180…for me’s is better to explain with the help of a table or figure What do you think?

Response 5: We strongly agreed with the reviewer’s comment, and explain it with the help of a table.

Page 6 Table 2 (The Table is displayed in WORD)

Point 6: Paragraph 3.4.2 .the inversion and validation in the area (Kenli District.) It’ better summarize all the results without repetitions, with patterns and tables and synthesizing. 

Response 6: Thanks for the reviewer's suggestion, and we have modified the way of summarizing the section Results and Analysis to eliminate repetitions as much as possible, especially Paragraph 3.4.2. I hope I have understood your suggestion correctly.

Point 7: In the conslusions it is advisable to define the results from future approaches and the consequences of the study 

Response 7: Thanks for the reviewer's reminding, and added the definition of the results from future approaches and the consequences of the study in the Discussion.

Page 13 Line 418-425:

    In contrast with previous studies, this paper integrates multi-sensors, multi-modeling methods, and multi-scale mapping and achieved high precision, high spatial and temporal resolution, and multi-scale inversion of soil salinity in the research area [61-63]. However, there are still three aspects that need to be further studied. First, the precision needs to be further improved, and contributions of different data sources and nonlinear modeling methods to the precision will continue to be explored [64,65]. At the same time, the driving factors and formation mechanism of soil salinization must be elucidated to prevent this problem [66,67]. Furthermore, the universality of time and space of monitoring should be improved [68,69].
